# Variant proteins stimulate more IgM+ GC B-cells revealing a mechanism of cross-reactive recognition by antibody memory

Bronwen R Burton[1†], Richard K Tennant[2†], John Love[2], Richard W Titball[2], David C Wraith[3], Harry N White[2*]

[1]Faculty of Biomedical Sciences, University of Bristol, Bristol, United Kingdom; [2]Department of Biosciences, University of Exeter, Exeter, United Kingdom; [3]Institute of Immunology and Immunotherapy, University of Birmingham, Birmingham, United Kingdom

**\*For correspondence:**
h.n.white@exeter.ac.uk

[†]These authors contributed equally to this work

**Competing interests:** The authors declare that no competing interests exist.

**Abstract** Vaccines induce memory B-cells that provide high affinity secondary antibody responses to identical antigens. Memory B-cells can also re-instigate affinity maturation, but how this happens against antigenic variants is poorly understood despite its potential impact on driving broadly protective immunity against pathogens such as Influenza and Dengue. We immunised mice sequentially with identical or variant Dengue-virus envelope proteins and analysed antibody and germinal-centre (GC) responses. Variant protein boosts induced GCs with a higher proportion of IgM+ B cells. The most variant protein re-stimulated GCs with the highest proportion of IgM+ cells with the most diverse, least mutated V-genes and with a slower but efficient serum antibody response. Recombinant antibodies from GC B-cells showed a higher affinity for the variant antigen than antibodies from a primary response, confirming a memory origin. This reveals a new process of antibody memory, that IgM memory cells with fewer mutations participate in secondary responses to variant antigens, demonstrating how the hierarchical structure of B-cell memory is used and indicating the potential and limits of cross-reactive antibody based immunity.
DOI: https://doi.org/10.7554/eLife.26832.001

## Introduction

Antibody-based immunity is underpinned by memory B-cells that have undergone antibody somatic hyper-mutation (SHM) and selection for improved antigen binding in germinal centres (GCs) (*MacLennan et al., 1997*). Re-challenge with the same antigen stimulates a rapid, higher affinity, secondary antibody response.

Protective immunity to highly mutable viruses, like Dengue and Influenza, can be induced by vaccination but the high level of variation often leads to immune escape (*Nabel and Fauci, 2010*), leading to a focus on generating vaccine responses against conserved antigenic regions (*Wu et al., 2010*; *Corti et al., 2011*; *Wang et al., 2015*).

Memory B-cells of IgM and IgG isotypes can also re-instigate GCs after secondary exposure (*Dogan et al., 2009*; *Pape et al., 2011*; *McHeyzer-Williams et al., 2015*), but how this happens against variant antigens is poorly understood despite its potential impact on driving the most broadly protective immunity.

Several studies suggest diversity in the memory B-cell population, showing that cells can express IgM or IgG (*Dogan et al., 2009*; *Pape et al., 2011*), be mutated or non-mutated (*Kaji et al., 2012*) and have low affinities (*Smith et al., 1997*), but still persist in GCs (*Kuraoka et al., 2016*).

It has long been speculated that this diversity may facilitate the recognition of antigenic variants (*Herzenberg et al., 1980*; *Pape et al., 2011*; *Kaji et al., 2012*) which could stimulate secondary

**eLife digest** Many devastating infectious diseases are caused by viruses that change over time. When a vaccine exists, it usually protects against a particular strain of virus, but often fails to defend against new versions of the microbe. This is why the flu vaccine has to be 'updated' every year, for example.

Vaccines rely on the memory of our immune system. When a virus enters the body, a group of immune cells known as B cells gets activated. Certain B cells can recognise the invader and produce specific proteins, the antibodies, which can target and kill the invader. During the infection some of these B cells become 'memory B cells', having gone through a maturation process that hones their ability to specifically recognize this particular microbe. If the same virus enters the organism again, the memory B cells rapidly identify it and produce a quicker and more efficient immune response than during the first attack. This is how vaccines work. However memory B cells may not be able to recognize a previous intruder if it has changed too much.

The memory B cell population is diverse. Some cells are fully mature and can quickly recognize the original virus. But others have not finished their maturation process: these cells are less focussed, and cannot target the original microbe with the same exact precision shown by mature memory cells. For almost forty years it was thought that this reduced focus might make the immature cells better at identifying new versions of the original attacker, but up until now, it was not clear what these memory cells could do.

Here Burton, Tennant et al. injected a group of mice with proteins from the Dengue virus, which prompted an immune reaction. After several weeks, the animals received either the same proteins again, or proteins that were different. Compared to the fully mature cells, the immature memory B cells were much better at recognizing the variants of the proteins, and these cells then multiplied and mounted an immune response. Without the original protein injection, the response without the immature memory B cells was not as efficient. The body therefore has a pool of memory B cells that can recognise a wider range of virus protein variants than the ones that caused the first immune reaction.

Understanding the role of immature memory B cells in immunity could help design vaccines that protect against several strains or fast-evolving viruses. This could have the potential to reduce the severity of diseases that affect hundreds of millions of people every year.

DOI: https://doi.org/10.7554/eLife.26832.002

GCs derived from less mutated, naïve-like, memory B-cells that still had an advantage over naive B-cells due to their increased numbers, pre-selected V-genes and lower activation thresholds (*Good and Tangye, 2007*; *Good et al., 2009*).

By sequentially immunizing mice with the same or different Dengue-virus envelope proteins, and analyzing serum antibodies and GC B-cells, we provide evidence that supports the hypothesis that less developed memory B-cells are used in secondary responses to variant antigens.

## Results

### E-protein variants elicit secondary serum antibody responses with different speed and cross-reactivity

We chose Dengue-3 envelope protein (E3) for all priming immunisations. Boost immunisations were performed 38 days later with identical E3 protein or variant E2 or E4 proteins which have 68% and 63% overall sequence identity with E3, respectively.

The cross reactivity of E3-primed mouse serum IgG correlated with sequence identity (*Figure 1A*), and overall cross-reactivity also correlated (*Figure 1B*).

Boosting with homotypic E3 antigen induced a rapid antibody memory response with anti-E3 titres rising rapidly to day 7, and not increasing further (*Figure 1D*). E-protein boosted antisera was not reactive with an irrelevant His-tagged protein (PR8 HA)(*Figure 1C*).

Heterotypic boosting with E2 induced a rapid and significant increase in anti-E3 titre, as might be expected if cross-reactive memory antibodies against the priming E3 antigen were recalled

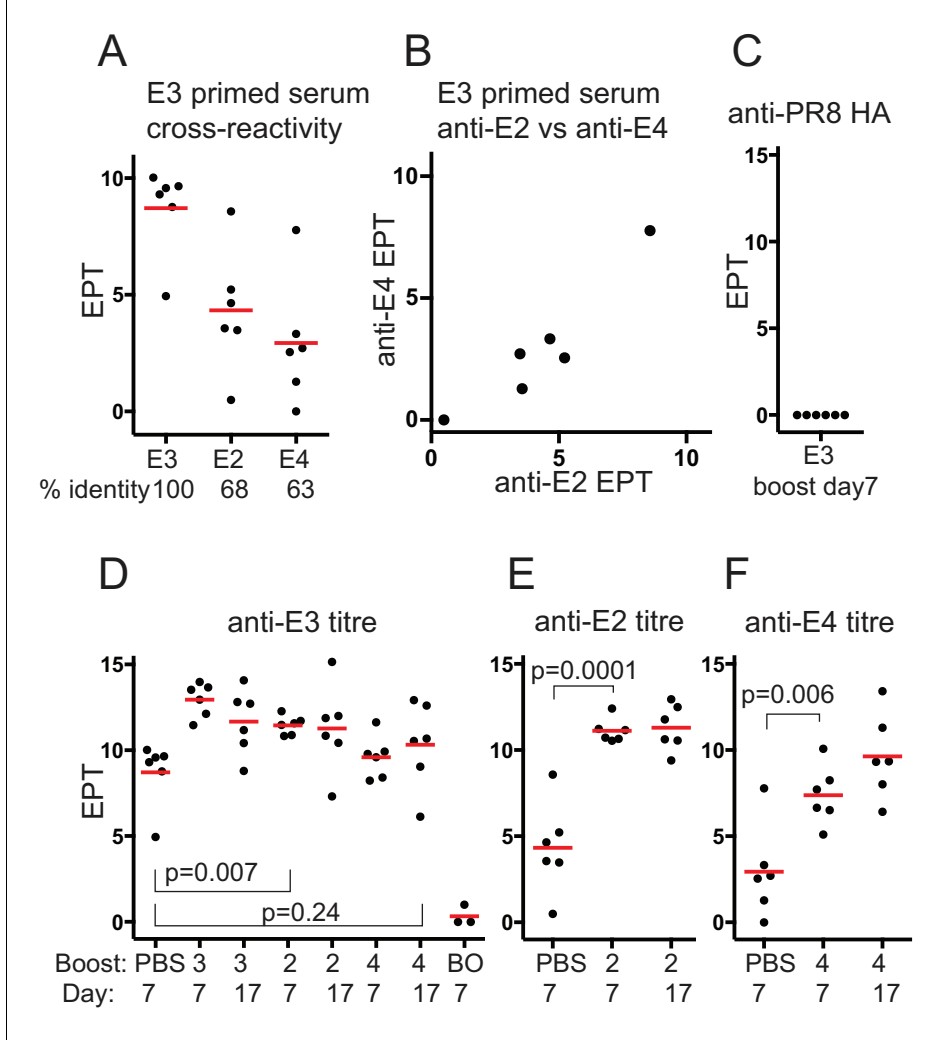

**Figure 1.** Serum antibody responses after boosting with Dengue envelope protein variants. (**A**) Cross-reactivity of E3 primed serum IgG with E-protein variants. Red bar shows mean value. Serum used was from mice mock-boosted with PBS 37 days after E3 priming and obtained 7 days later; E3, Dengue-3 envelope protein; E2, Dengue-2 envelope protein; E4, Dengue-4 envelope protein; % identity, sequence identity between E3 envelope protein and respective protein; end-point titre (EPT) values plotted are log2 of 1/(end point dilution x 100), each unit increase represents a doubling of titre. (**B**) E3 primed mouse serum cross-reactivity with E2 versus E4. (**C**) Control. Anti-PR8 HA serum IgG titre of E3 day 7boost serum. (**D**) Anti-E3 serum IgG titre after boosting with respective proteins. Red bar shows mean value. n = 6 from two independent experiments for each group except boost only, n = 3; first set of data points reproduced from panel A for comparison; numbers 3, 2 and 4 refer to serotype of Dengue-envelope protein used for boost; BO, adjuvant primed, E3 boosted, analysed 7 days later; Day, days after boosting. p-values calculated using two-tailed Students t-test after testing for equality of variance. (**E**) Anti-E2 serum IgG titre after E2 boost. Red bar shows mean value. n = 6 from two independent experiments for each group; labeling and statistics as for panel D. (**F**) Anti-E4 serum IgG titre after E4 boost. Red bar shows mean value. n = 6 from two independent experiments for each group; labeling and statistics as for panel D.

DOI: https://doi.org/10.7554/eLife.26832.003

(*Figure 1D*), that did not increase further by day 17. E4 boosting induced a modest but not statistically significant increase in the anti-E3 titre, even by day 17, showing the E4 variant boost had not induced a significant anti-E3 antibody memory response, or the induced antibodies had a low affinity for E3 (see discussion).

The anti-E2 titre induced by the E2 boost increased about 120-fold by day 7 (*Figure 1E*), and did not increase further by day 17, further indicating that E2 boosting induced a rapid memory-like serum IgG response against E2 derived from cross-reactive E3 primed memory B-cells. Conversely the anti-E4 titre, induced by E4 boosting, rose significantly but to a lower level, about 20-fold, by day 7 (*Figure 1F*) and showed a further rise by day 17. A boost alone did not induce a detectable antibody titre however, ('BO', *Figure 1D*) suggesting a role for memory B-cells of some type and/or cross-reactive T-cell memory, facilitating the E4 boost response.

## Increased levels of IgM+ GC B-cells with fewer mutations after variant protein boosting

E3 and E2 boosting induced early GC B-cell levels similarly by day 7, to 4.5–5.5% of total lymphocytes, which then reduced by two-thirds by day 17 (*Figure 2B*). E4 boosting induced GC B-cell levels about a third as high, which then reduced similarly by about 60% at day 17, remaining 4-fold higher than controls.

Analysis of the proportion of IgM+ GC B-cells showed a highly significant trend at day 7 after boosting, with the proportion of IgM+ GC B-cells correlating with increasingly variant challenge (*Figure 2C*). This trend continued to day 17. The proportion of IgM+ B cells was also consistent between individuals in an experimental group (*Figure 2D*).

Overall levels of VH mutations increased in all groups from day 7 to day 17 (*Figure 2E*), consistent with secondary affinity maturation. Sequences are available in *Supplementary file 1*.

There were lower levels of SHM in IgM+ GC B-cells 7 days after the variant boosts, particularly with the most variant protein E4, compared to the homotypic E3 boost (*Figure 2F*). Boosting with variant proteins, therefore, induced early GCs with increased proportions of IgM B-cells that had fewer VH mutations.

Analysis of the VH clonality of GC B-cells after E-protein boosts showed that almost every VH sequence was from a distinct B-cell clone (*Figure 2G*). These data also showed that the two variant boosts elicited different repertoires of VH. 40% of the VH sequences sampled at day 7 from E2 boosted mice were either VH14-3 or the closely related VH14-4 (black dots, *Figure 2G*), suggestive of a secondary response more focused on a particular epitope (see discussion). Some of these VH were also present in the homotypic E3 boost day 7 samples, but neither were detected at day 7 after E4 boosting (*Figure 2G*).

## Changes in serum affinity/avidity after variant antigen boosting

E2-variant boosting induced an immediate and significant increase in avidity by day 7 (*Figure 3A*) which did not detectably change until perhaps day 32, although data variability is high. A modest but significant increase in serum affinity, however, was detected by day 17, with a further increase detected by day 32 (*Figure 3C*). We interpret this to mean that a relatively small portion of serum IgG underwent affinity maturation by day 17 in response to the E2 boost and was not detectable by the Urea avidity assay due to high variability and the high pre-existing IgG titres (*Figure 1E*), or other limitations of the Urea assay (*Alexander et al., 2015*). Boosting with the E4 variant elicited slower increases in relative affinity and avidity, only detectable by day 32, but by then representing an equivalent, if not greater, increase compared to that induced by E2 (*Figure 3B and D*).

## Similar memory T-cell stimulation by variant dengue E-proteins

Memory T-cells are necessary for memory B-cell responses against haptens and viral proteins (*Aiba et al., 2010*; *Hebeis et al., 2004*). We found no evidence that the memory T-cell response to re-stimulation by variant E-proteins was any different from re-stimulation by E3 (*Figure 3E*). These data imply that a deficiency in T-cell recognition of these antigens cannot explain the differences in response to E2 and E4 challenge, and supports the idea that either T-cell receptors can recognize antigenic peptides from regions with around 50% sequence difference (see discussion) or, more likely, B-cells present peptides from different, more conserved regions than those their antibodies bind to.

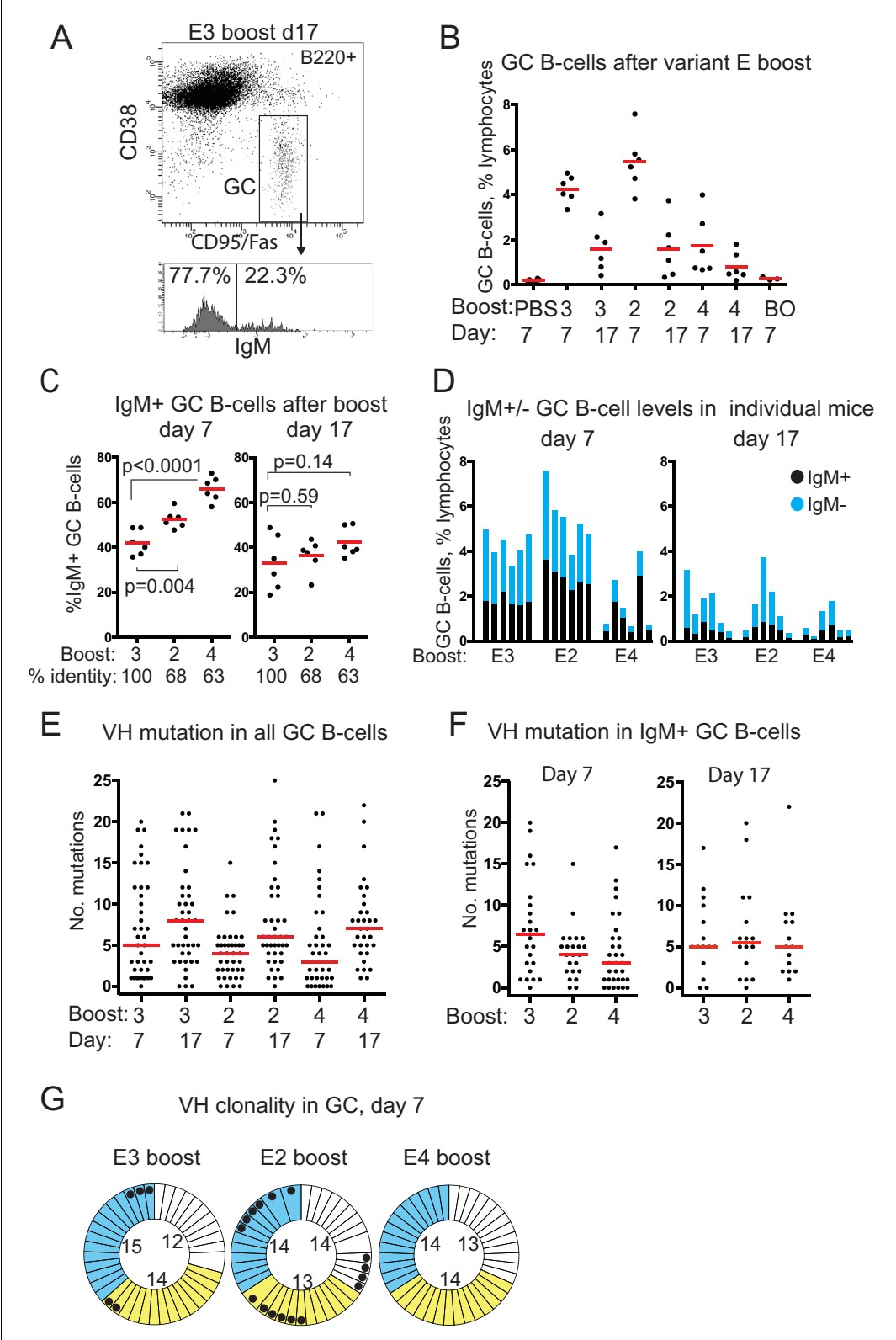

**Figure 2.** GC B-cell levels, isotypes, VH mutation and clonality after boosting with E-protein variants. (**A**) FACS gating strategy used to identify and sort GC B-cells and determine isotype. (**B**) GC B-cell levels after E-variant boosting, expressed as % total lymphocytes; Red bar shows mean value; numbers 3,2 and 4 refer to serotype of Dengue-envelope protein used for boost; BO, boost only, adjuvant primed, E3 boosted day 37, analysed 7 days later; Day, days after boosting. (**C**) % IgM+ GC B-cells, of total GC B-cells, after boosting. Red bar shows mean value. n = 6 from two independent
*Figure 2 continued on next page*

*Figure 2 continued*

experiments for each group; labels as for panel B except % identity which refers to sequence identity between E3 and other variants; p-values calculated using two-tailed Students t-test after testing for equality of variance. (D) Levels of IgM+ and IgM- GC B-cells in individual boosted mice. (E) Number of mutations detected in VH of all isotypes of GC B-cells, from n = 3 mice except E4 boost day 17, n = 2. Red bar is median value. VH region sequenced is CDR1 to FR3; labeling as panel B. (F) Number of mutations detected in VH of IgM+ GC B-cells, from n = 3 mice except E4 boost day 17, n = 2. Red bar is median value. (G) Clonality of sequences from single GC B-cells 7 days after boosting; colours indicate different mice in each group; thin sectors, unique sequences; thicker sectors two or three clonal sequences according to sector size; black dots, VH 14–3 or VH14-4 sequences; numbers in circles, number of sequences from that mouse; Identical VH clones had the same: V-gene, CDR3 length, J-gene, D-gene if assigned, D-reading frame, three or fewer differences in CDR3 amino acid sequence.

DOI: https://doi.org/10.7554/eLife.26832.004

## The primary antibody and GC response to E4

For comparison with the E4 boost response, we performed primary immunisations with E4 and analysed serum antibodies and GC B-cells at day 7 and day 17. Serum levels of anti-E4 IgG rose to a moderate level by day 17 (mean EPT = 3.6, *Figure 4A*), being less than seen after E4 boosting (*Figure 1F*). GC B-cell levels rose to a mean of 0.8% lymphocytes at day 7 after E4 priming, half as much as after the E4 boost, then fell similarly to the post boost samples by around 60% by day 17 (*Figure 4B*). As with the E-boost GCs, the proportion of IgM+ GC B-cells fell over time (*Figure 4C*) and levels of VH mutation in all B-cells and IgM+ B cells increased (*Figure 4D and E*). The median level of VH mutation in IgM+ GC B-cells at day 7 after E4 priming is less (=2) than after E4 boosting (=3) suggesting, not conclusively, that GC Bells at day 7 after E4 boosting are memory derived. Antibody titres were insufficient to do a relative affinity competition ELISA and no 7M Urea-resistant IgG was detected 7 or 17 days after E4 priming (data not shown).

## IgM antibodies from E4 boost GC show evidence of prior selection

If E4 boost induced B-cells are memory derived the antibodies should show evidence of pre-selection by the E3 prime. We made 48 recombinant antibodies (rAbs), 38 of which were IgM (*supplementary file 2*), 24 from E4 primed mice (day 7 and day 17) and 24 from E4 boosted mice (day 7). *Figure 4F* and *Supplementary file 2*, show the results from the initial screen of all rAbs against E4, indicating that the efficiency of detection of positive binding (deemed as O.D. > 0.1, useful for subsequent titration) was quite low but consistent with the 30–50% binding frequency of GC rAbs previously observed (*Kuraoka et al., 2016*), except for E4 prime day 7, which has only 2/13 rAbs binding strongly enough to be titrated. This might be expected of antibodies from a day 7 primary response GC, and indicated they were overall of lower affinity. Other rAbs from this group showed evidence of weak binding (*supplementary file 2*), indicating that the rAb cloning efficiency for this group was not reduced and only the two strongest binders were above the ELISA titration threshold. All but one of the positive binding rAbs were IgM. *Figure 4G* shows the ELISA titration and *Figure 4H* the derived endpoint titres, which we are using as a proxy of affinity. A more strongly binding IgM rAb from E4 boost day 7, B5, and the only positive binding IgG1 rAb, G6, are indicated on *Figure 4H*. The positive-binding rAbs from E4 prime day 17 show a higher affinity than those from prime day 7, consistent with affinity maturation. Six of the seven positive-binding IgM rAbs from E4 boost day 7 show a higher affinity than the two strongest binding IgM rAbs from E4 prime day 7. This is consistent with pre-selection by the E3 prime immunization, and also considering the higher proportion of rAbs with an anti-E4 O.D. > 0.1, implies the GC B-cells expressing these antibodies are memory derived. rAb affinities were generally low, which might be expected of IgMs particularly in early GCs. We estimated the Kd of rAbs B5 and G6 (an IgG1) as around 150 nm and 1 μm respectively (see Materials and methods). Other rAbs would be in the super-micromolar range. *Figure 4I* shows the cross reactivity of rAbs with E3. Binding to E3 correlates with binding to E4, but because of the generally low rAb affinities we suggest that the antibodies cannot discriminate between similar epitopes. The higher affinity of E4 boost rAbs B5 and G6, and binding to E3, suggest they may have genuine specificity for E3, thus consistent with their derivation from anti-E3 memory. That rAb B5 is an IgM with only one VH (and one Vkappa) mutation, provides further support for the proposal of this study.

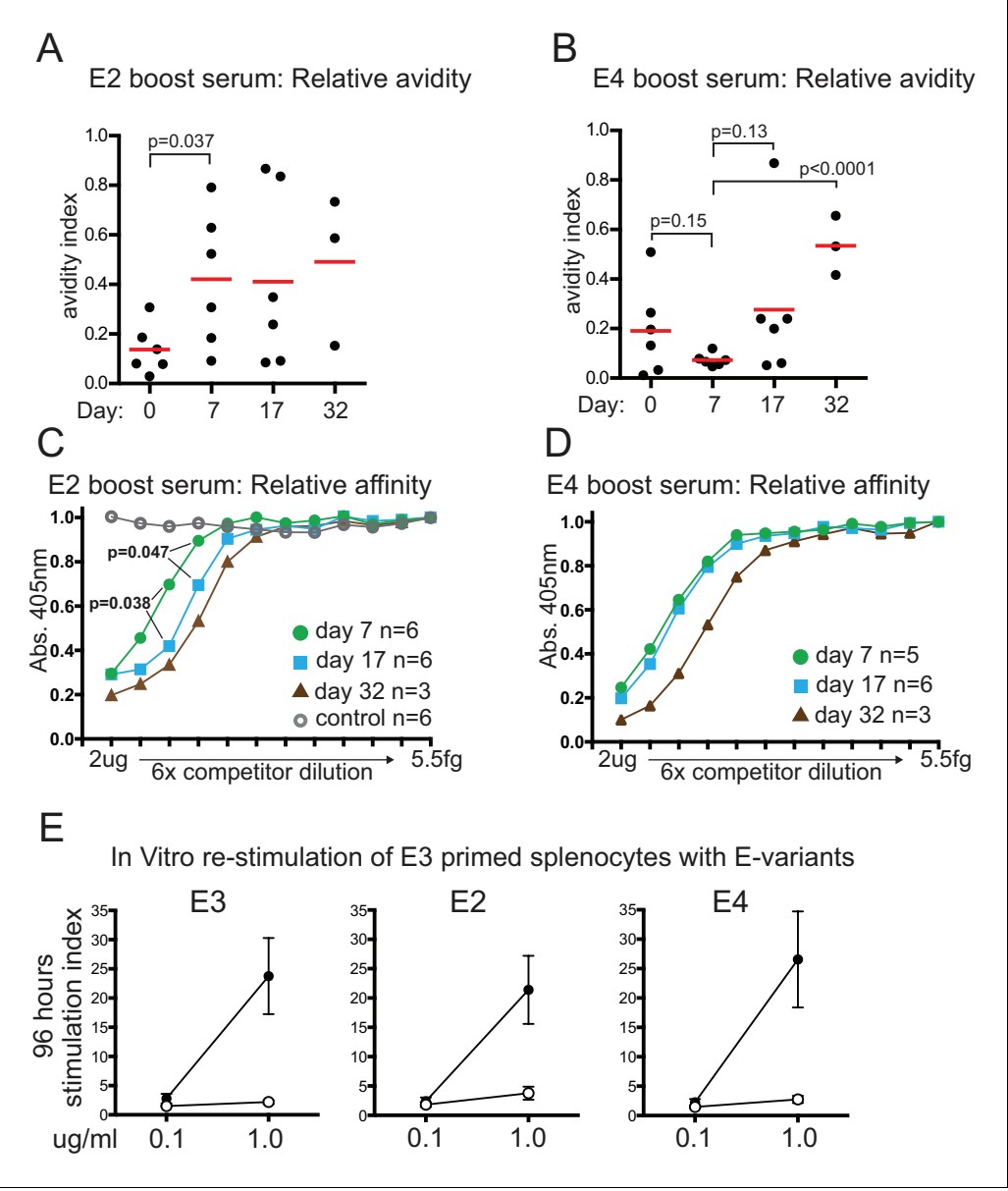

**Figure 3.** Relative serum affinity and avidity after boosting with E-protein variants, and T-cell re-stimulation. (**A**) Relative avidity of E2 boost serum for E2, measured by resistance to 7M Urea. Red bar shows mean value; Day, days after E2 boosting; Day 0 sample was from mice mock-boosted with PBS 37 days after priming with E3 and obtained 7 days later. (**B**) Relative avidity of E4 boost serum for E4, measured by resistance to 7M Urea. Labeling as for panel A; Day 0 sample was from mice mock-boosted with PBS 37 days after priming with E3 and obtained 7 days later (**C**) Relative affinity of E2 boosted serum for E2. Inhibition by lower concentration of competitor implies higher affinity of serum for competitor. Maximum competitor amount 2 µg in 50 µl followed by six-fold dilutions of competitor; timepoint of samples and numbers of individuals in group indicated. Open circles, E2 boost day 17 serum competed with irrelevant His-tagged protein measured on E2 target (**D**) Relative affinity of E4 boosted serum for E4. Labeling as for panel A. (**E**) T-cell proliferation measured by $^{3}$H incorporation 96 hr after re-stimulation in vitro with indicated amounts of E-protein variants; error bars indicate standard error of the mean; n = 4 or five from two independent experiments (see source data). Closed symbols, E3 primed mouse splenocytes re-stimulated with indicated E-protein variant. Open symbols, adjuvant primed mouse splenocytes re-stimulated with indicated E-protein variant.

DOI: https://doi.org/10.7554/eLife.26832.005

The following source data is available for figure 3:

**Source data 1** Source data for *Figure 3* panels C, D and E.

DOI: https://doi.org/10.7554/eLife.26832.006

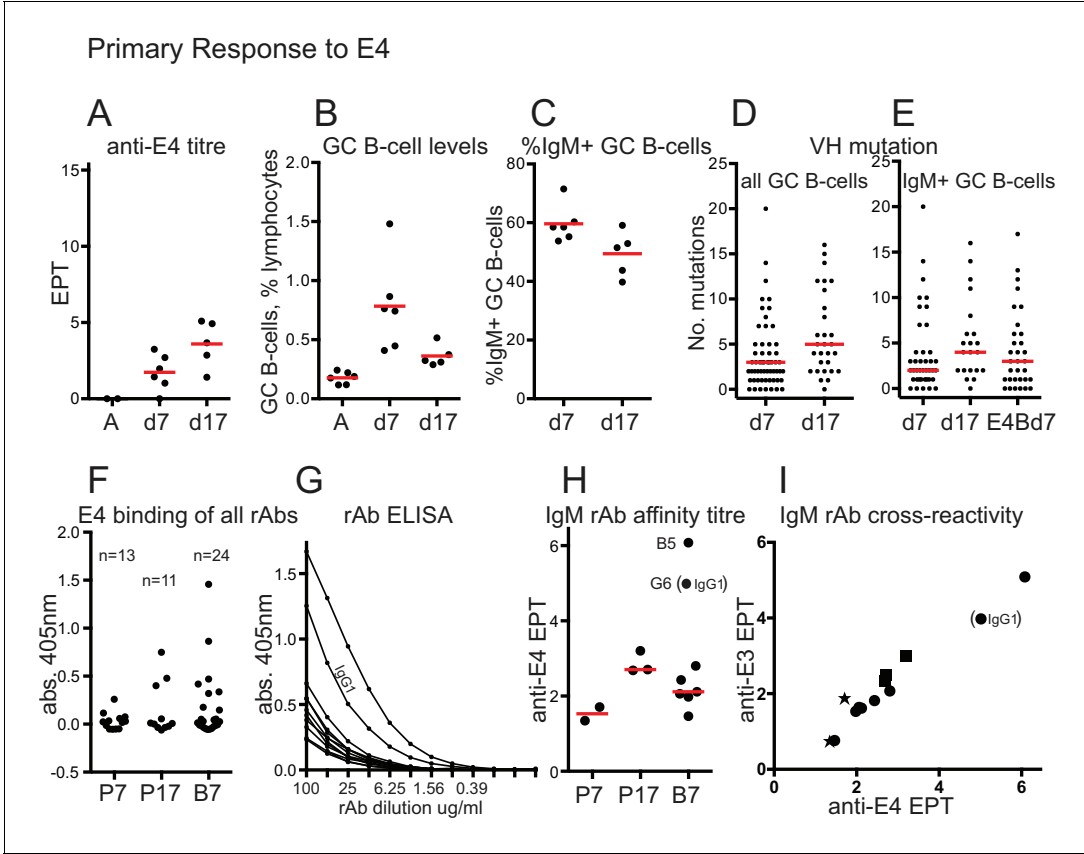

**Figure 4.** Primary response to E4 and rAb binding. (**A**) anti-E4 IgG titre after E4 priming; Red bars show mean titres; A, serum from adjuvant-only primed mice at day 45; d7, 7 days after E4 priming; d17, 17 days after E4 priming; EPT, end-point titre calculated as for *Figure 1*. (**B**) GC B-cell levels after E4 priming; Red bars indicate mean levels; A, cells from adjuvant-only primed mice 7 days after priming; other x-axis labels as for panel A. (**C**) % IgM + GC B-cells after E4 priming; Red bars show mean values; x-axis labels as for panel A. (**D**) Numbers of VH mutations in all isotypes of GC B-cells after E4 priming; Red bars show median values, from n = 3 mice (d7) and n = 2 mice (d17); x-axis labels as for panel A. (**E**) Numbers of VH mutations in IgM+ GC B-cells after E4 priming and boosting; Red bars show median values, from n = 3 mice (d7), n = 2 mice (d17) and n = 3 mice E4Bd7; x-axis labels as for panel A except E4Bd7, 7 days after E4 boosting which was 38 days after E3 priming. (**F**) ELISA screen of binding of all 48 rAbs. rAbs incubated at 100µgml$^{-1}$. Number of rAbs in each group indicated. P7, 7 days after E4 prime; P17, 17 days after E4 prime; B7, 7 days after E4 boost. As the antibodies were cloned as chimeric human IgG1 antibodies the background from non-specific human polyclonal IgG binding has been subtracted from O.D. readings. Values in *supplementary file 2*. (**G**) ELISA titration of rAbs that showed binding O.D. > 0.1 in panel F. All but one were IgM. IgG1 rAb indicated. Background subtraction as for panel F, using appropriate dilution of polyclonal IgG. (**H**) Anti-E4 end point titre of positive-binding rAbs, used as a proxy of rAB affinity. X-axis labels as for panel F. End-point titre values plotted are log2 of 1/end point dilution (undiluted = 100µgml$^{-1}$). Red bars show median values (excluding any IgG1 data). Stronger binding IgM rAb 'B5', and IgG1 rAb 'G6' EPT readings indicated. (**I**) anti-E3 versus anti-E4 endpoint titres. Star, E4 prime day 7 rAbs; Square, E4 prime day 17 rAbs; circle, E4 boost day 7 rAbs. IgG1 EPT reading indicated. End-point titre values plotted are log2 of 1/end point dilution (undiluted = 100µgml$^{-1}$).

DOI: https://doi.org/10.7554/eLife.26832.007

## Discussion

The most variant protein we boosted with, E4, stimulated GCs with the highest proportion of IgM + cells and with the lowest levels of VH gene mutation, greater VH-gene diversity, and a slower, more specific, serum IgG response that resulted in equivalent if not higher affinity, compared to the heterotypic E2 boost. This response was higher than the primary response to E4. IgM rAbs cloned from E4 boost day 7 GC showed a higher affinity for E4 than those from E4 primed day 7 GC, implying they were memory derived. This demonstrates that IgM memory cells with fewer mutations, from 'lower' levels of the memory compartment, participate in secondary responses to variant antigens, and further challenges the hypothesis that highly mutated, class-switched cells elicited by homotypic antigen boosting are a 'mirror' of the antibody memory compartment (*Weiss and Rajewsky, 1990*). The slower nature of the E4 boost serum response also suggests a lower level of

immediate differentiation of memory cells into AFCs than seen with for example the homotypic or E2 response, and is consistent with reduced numbers of high affinity class-switched memory cells recognizing E4.

The serum antibody response to the closer variant, E2, was more rapid, more cross-reactive and evidenced some earlier affinity maturation. These observations are consistent with a response derived more from the 'higher' layers of the E3 specific memory compartment. The IgM +cells induced by E2 boosting have more mutations than after E4 boosting, indicating they are memory derived. As there are higher proportions of these IgM+ GC B-cells, with fewer mutations relative to the homotypic E3 boost, this provides further support for the hypothesis that IgM+ B cells with fewer mutations furnish memory responses to variant antigens

Naïve B-cells may contribute to the IgM+ GC B-cells we observe after E4 boosting, although the higher affinities of the rAbs from this group suggest many are memory derived. Also, the slightly higher median level of VH mutation and the higher levels of IgM+ GC B-cells after E4 boosting (2x) compared to priming, suggest IgM +memory B-cells are involved in the boost response consistent with the well established presence of IgM +memory cells with few or no mutations (*Dogan et al., 2009*; *Pape et al., 2011*; *Kaji et al., 2012*) and the known lower activation threshold of memory B-cells in response to antigen (*Good and Tangye, 2007*).

Whilst E3 specific memory cells may be expected to increase the anti-E3 titre when stimulated by a cross-reactive E4 boost, the small but not significant effect we observe (*Figure 1D*) is consistent with the lowest affinity, least mutated, E3-specific memory cells being stimulated by an E4 boost. Antibodies from such cells may, therefore not add much to the already high, affinity matured, anti-E3 titre induced by E3 priming. The 14-fold higher anti-E4 titre at day 7 after boost (*Figure 1F*) versus day 17 after prime (*Figure 4A*) also argues for a significant contribution from B-cell memory.

The fusion-loop epitope in domain 2 of the dengue envelope protein is 100% conserved between strains and in humans, antibodies against this are prevalent in cross-reactive secondary responses (*Lai et al., 2013*; *Chaudhury et al., 2017*). The E2 boost response is consistent with this effect, especially considering the restricted clonality seen in VH sequences, but the low anti-E3 titre induced by E4 is not. A recent study (*Chaudhury et al., 2017*) showed that the mouse response to recombinant E-protein is predominantly focused on domain 3 of the protein, and so cross reactivity with the fusion loop epitope (domain 2) should be less dominant. While E2 and E4 are 68% and 63% overall identical to E3, in domain 3, a focus of mouse antibodies, they are 62% and 51% identical, a bigger difference in differences, helping explain the responses we observe here.

## Materials and methods

### Animals, immunisations and antigens

Female 8–11 week old BALB/c mice were purchased from Charles River, U.K. Primary immunisations were intra-peritoneal (IP) with 25 µg recombinant Dengue envelope protein (Biorbyt, UK) precipitated in alum with $2 \times 10^7$ heat-killed *B.pertussis*. Secondary immunisations were IP with 25 µg recombinant Dengue envelope protein (Biorbyt) dissolved in phosphate-buffered saline (PBS). At designated time points mice were anaesthetized and bled for collection of serum and then humanely sacrificed for collection of spleen cells. Dengue envelope (E) proteins were C-terminal His-tagged and expressed in *E-coli* prior to purification. Dengue proteins were tested for endotoxin by LAL assay (Fisher Scientific, UK) and contained it at a low level: E2, 5.4EU/µg; E3, 2.5EU/µg; E4, 3.1EU/µg. Endotoxin in this range does not give a detectable physiological response in mice (*Copeland et al., 2005*).

### ELISA for serum and rAbs

ELISA plates (Nunc Maxisorp, Fisher Scientific, UK) were coated overnight at 4°C with 1 µg/ml protein in 0.1M bicarbonate buffer pH 9.3. Plates were washed three times in PBS/0.05% Tween-20 (Sigma, UK) (PBST) and blocked for 30mins at room temperature with PBST/2% bovine serum albumin (BSA, Sigma). Plates were then washed three times and incubated with serum dilutions in PBST/1.0% BSA for two hours at room temperature. After three washes plates were incubated with alkaline-phosphatase conjugated goat anti-mouse IgG (Sigma) for one hour at room-temperature, washed three times and developed with pNPP substrate (Sigma) for one hour. Absorbance was

measured at 405 nm. For the initial rAb screen, rAbs were incubated at 100μgml$^{-1}$ in PBST/1.0% BSA for 2 hr at room temperature on plates coated with E4 and blocked as above, and subsequently treated as above except with use of anti-human IgG second layer (Sigma). Background binding to plates was determined using binding of non-specific polyclonal human IgG at 100μgml$^{-1}$, because the rAbs were expressed as chimeric constructs with human constant regions, and this was subtracted from the rAb O.D. Positive binding rAbs were deemed to be those with O.D. > 0.1 that could be subject to an ELISA endpoint titration. For the ELISA titration and endpoint analysis, doubling dilutions of positive binding rAbs, and polyclonal IgG background subtraction control, were used starting at 100μgml$^{-1}$. Endpoint titre was set at O.D. = 0.1 and calculated using interpolation on Graphpad Prism. The assay was repeated using E3 coated plates to determine the rAB cross reactivity. The affinity (Kd) of rAbs B5 and G6 (the two strongest binding rAbs) was estimated from the inflection point of the ELISA titration curve as indicating 50% maximal binding, and on the assumption that at these higher antibody concentrations binding of rAB to immobilized antigen will have a minor effect on concentration of unbound rAb. We estimated the B5 inflection point to be at approximately 25ugml$^{-1}$ (=approx. 150 nM) and the G6 inflection point to be just above 100ugml$^{-1}$ (=approx. 1 uM)

## Competition ELISA

ELISA plates were coated as above with target protein, then washed, blocked and washed as above except the blocking was done at 37°C for one hour. Mouse serum samples were diluted in PBST/1% BSA to twice the concentration of the maximum dilution that gave an absorbance at 405nm = 1.0 in ELISA to the target protein. Serial six-fold dilutions of competitor protein were made in PBST/1% BSA, such that the highest concentration of competitor was 2.4 μg in 30 μl. 30 μl of diluted serum was mixed with 30 μl of each competitor protein dilution and incubated in a polypropylene 96-well plate at 37°C for 1 hr. Serum/competing antigen mixture (50 μl) was then added to each well of the target antigen coated plate and incubated at 37°C for one hour. Plates were washed as above and then 50 μl of alkaline–phosphatase conjugated anti-mouse IgG (Sigma) was added to each well followed by incubation at 37°C for one hour. Plates were washed as above and incubated with 75 μl per well of p-nitrophenyl phosphate substrate (Sigma) for one hour at room temperature. Absorbance was measured at 405 nm. All individual serum dilutions were also reacted in the absence of competitor, against BSA coated wells, following the same incubation protocol. These background values were subtracted from the competition ELISA values obtained above. The readings were then normalized so that the samples with the maximum competitor dilution gave a value of 1.0

## Urea avidity ELISA

Adapted from *Puschnik et al., 2013*. Assay plates were coated with antigen and blocked as for the ELISA protocol. 1/200 dilutions of serum in PBST/1% BSA were incubated on plates for 2 hr at room temperature. Wells were washed once with PBST, incubated for 10 min at room temperature with PBST or PBST/7M Urea, washed a further two times with PBST and then treated as for standard ELISA. The avidity index was calculated by dividing readings from 7M Urea treatment by readings from PBST-only treatment, after subtraction of background absorbance.

## Flow cytometry

Whole spleen cell-suspensions were red-cell depleted with Pharm-Lyse (BD Biosciences, UK) and incubated with anti-CD16/32 monoclonal antibody (Fc-block, BD Biosciences) for 15 min at 4°C. Cells were then stained with APC anti-B220, BV421 anti-CD38, PE anti-CD95/Fas (all BD) and FITC anti-IgM (eBioscience, Thermofisher Scientific, UK) for 45 min at 4°C. After washing, cells were re-suspended in PBS 5% FCS (Gibco, Thermofisher Scientific) and analysed or single-cell sorted on a FACS Aria II (BD).

## GC B-cell antibody sequencing, cloning, expression and purification

Single GC B-cells were sorted into half a 96 well PCR plate (less three control wells) containing10μl of chilled 10 mM Tris pH 8.0, 1 U/μl RNAsin (Promega, UK) and placed on dry ice then at −80°C. One-Step RT-PCR (Qiagen, UK) was performed according to manufacturers instructions, by adding 15 μl RT-PCR master mix, using first-round primer sets described in *Tiller et al. (2009)*, with heavy-

chain and kappa-chain primers, for 50 cycles, annealing at 53.6°C. Heavy-chain second-round PCRs were performed using 2 µl first-round product and the nested/semi-nested primer sets from *Tiller et al. (2009)*, with Hot Star Taq polymerase (Qiagen) for 50 cycles annealing at 56°C. Second round PCR product (4 µl) was analysed on a 1.2% agarose gel. Successful PCRs were then Sanger sequenced. For this study the sequencing primer was the pan VH primer 5'MsVHE (*Tiller et al., 2009*) which leaves part of the 5' of FR1 unsequenced. For this reason the FR1 sequence was not included in the analysis. VH sequence identification and SHM analysis was done using the IMGT V-Quest online platform. VH sequences are in *Supplementary file 1*. Further cloning, construction and expression of antibodies as chimeric IgG1 rAbs was done according to *Tiller et al. (2009)*. Briefly, second round PCRs of in-frame VH and VK sequences were repeated with V-gene specific primers that included a restriction site for sub cloning (*Tiller et al., 2009*). These PCR products were purified (Qiagen), restriction digested, purified (Qiagen) and ligated (instant sticky-end ligase, NEB, UK) into the appropriate expression vector containing either human IgG1 or Kappa constant regions, prior to transformation into *E. Coli* NEB5-alpha (NEB). Expression constructs in transformed colonies were verified by sequence analysis prior to preparation of plasmid mini-preps (Qiagen). 293A cells were split and grown to 80% confluence in DMEM with ultra-low IgG FCS (PAN Biotech, Germany) in 150 mm plates prior to replacement of medium with 20 ml Panserin 293A serum free medium(PAN Biotech). 15 ug each of matched VH and VKappa constructs were added to 2 ml saline with 90 ug PEI, briefly vortexed and rested for 10mins. Transfection solution was added to plates and mixed gently. After 3 days medium was collected, centrifuged at 800 g for 10mins to clear debris, and further medium added. After a further 3 days medium was collected, cleared of debris as before and pooled. 100 ul protein-G sepharose (GE Healthcare, UK) was added to super-natants and incubated with rocking overnight at 4°C. Protein G sepharose was collected by centrifugation at 800 g for 10 mins and transferred in PBS to a PBS equilibrated spin column (Bio-Rad, UK). After 3 rounds of washing with 800 ul of PBS, rAbs were eluted in two 200 ul passes of 0.1M Glycine (pH2.9) into a tube with 40 ul of 1M Tris pH 8.0, 0.5% Sodium Azide. Antibody concentrations were determined by O.D. on a Nanodrop instrument (Thermo) and corrected for an extinction co-efficient of 1.36.

## T-cell proliferation assay

Spleens were harvested from female BALB/c AnCrl mice 39 days after challenge. Splenocytes ($5 \times 10^5$) were cultured in triplicate with the indicated concentration of E-protein in X-VIVO 15 medium. Cells were cultured for 96 hr and 0.5 µCi of [$^3$H] thymidine was added to wells for 16 hr before measurement with a 1450 MicroBeta counter (Wallac, Perkin Elmer, UK).

## Statistics

For statistical analysis sample sizes were chosen to address group size reductions that observe the ARRIVE guidelines. Cages of three mice were randomly allocated to treatment groups. These group treatments were independently biologically replicated to give a sample size of 6. Where statistical analysis was applied, data points were analysed with Levene's test for equality of variance and where violated they were subject to a two-tailed Students t-test for unequal variance, otherwise the two-tailed t-test for equal variance.

## Acknowledgements

We are especially grateful to the late Michael Neuberger for critical discussion and early comment on the project. Thanks to Patrick Wilson and Christian Busse for advice on single cell antibody PCR, Per Klasse for advice on antibody avidity assays, James Cresswell for advice on statistics, Kai Toellner for discussions and Jamie Gilman for extra FACS work.

## Additional information

### Funding

| Funder | Grant reference number | Author |
|---|---|---|
| Wellcome Trust | 100115/Z/12/Z | Harry N White |

The funders had no role in study design, data collection and interpretation, or the decision to submit the work for publication.

### Author contributions

Bronwen R Burton, Richard K Tennant, Data curation, Formal analysis, Investigation, Visualization; John Love, Resources, Writing—review and editing, Approved final version and revision; Richard W Titball, Resources, Writing—review and editing; David C Wraith, Conceptualization, Resources, Supervision; Harry N White, Conceptualization, Data curation, Supervision, Funding acquisition, Validation, Investigation, Visualization, Methodology, Writing—original draft, Project administration, Writing—review and editing

### Author ORCIDs

Richard K Tennant (iD) http://orcid.org/0000-0003-3033-1858
David C Wraith (iD) http://orcid.org/0000-0003-2147-5614
Harry N White (iD) http://orcid.org/0000-0001-8186-7789

### Ethics

Animal experimentation: All animal experiments were done following the ARRIVE guidelines under authority of UK Home Office license PPL 30/3089, with permission from University of Exeter, UK, local animal welfare ethical review board. Blood samples were taken under Ketamine and Xylazine anaesthesia and every effort was made to minimise suffering.

### Decision letter and Author response

Decision letter https://doi.org/10.7554/eLife.26832.012
Author response https://doi.org/10.7554/eLife.26832.013

## Additional files

### Supplementary files

• Supplementary file 1. GC B-cell VH Sequences. VH sequences from single sorted GC B-cells. Sequences are grouped into treatment groups, and within this, arranged in blocks for sequences from individual mice. Raw sequences were analysed by IMGT V-Quest. Due to cloning and sequencing primers being at start of FR1 region, this region not included in mutation analysis. CDR1T, total mutations in CDR1; CDR1S, silent mutations in CDR1; CDR1R, replacement mutations in CDR1; likewise for FR2, CDR2 and FR3 regions; Tot Mut, total mutations in CDR1 to FR3 regions.
DOI: https://doi.org/10.7554/eLife.26832.008

• Supplementary file 2. Data on recombinant antibodies
DOI: https://doi.org/10.7554/eLife.26832.009

• Transparent reporting form
DOI: https://doi.org/10.7554/eLife.26832.010

### Data availability

All data generated or analysed during this study are included in the manuscript and supporting files. Source data files have been provided where only average values are plotted.

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
