## [Decision Letter]

Thank you for submitting your article "Variant proteins stimulate more IgM+ GC B-cells revealing a mechanism of cross-reactive recognition by antibody memory" for consideration by *eLife*. Your article has been reviewed by three peer reviewers, one of whom, Arup Chakraborty, is a member of our Board of Reviewing Editors and also acted as the Senior Editor.

The reviewers have discussed the reviews with one another and the Reviewing Editor has drafted this decision to help you prepare a revised submission.

Summary:

In this paper, the authors study whether cross-reactive memory B cells re-instigate GCs upon sequential immunization with variant antigens. They also study which types of memory B cells initiate such GCs. Toward this end, they prime with a particular strain of dengue virus envelope protein (E3) and then boost with either one of two variants, E2 and E4. E2 has 68% sequence identity with E3 while E4 has 63% . Memory B cell cross reactivity has been previously discussed by several groups, but the present study could be valuable in showing its existence in virus polyepitopic antigens, rather than the monoepitope hapten-carrier system.

The main finding is that variant proteins (dengue virus) tend to induce germinal centers (GCs) with higher levels of IgM+ memory B cells than the wild type immunizing protein in a mouse model. The most variant protein induced the highest proportion of IgM+ memory cells and a slower IgG response. These findings are potentially interesting and valuable. However, some significant concerns that need to be addressed are noted below. Some of these points involve making clarifications or elaboration of existing data, and only a few new experiments are required.

Essential points:

1) One major criticism is that the authors are not able to determine whether the IgM+ GC cells are derived from activated naive B cells or from IgM memory B cells. They also do not clone any mAbs from the GC B cells to analyze binding cross-reactivity (only sequence data is shown). The paper would be considerably stronger with this data.

2) Figure 1 D and F: The authors show that boosting with E4 did not significantly increase the serum titer to E3 at day 7 or 14. However, the anti-E4 titer rose significantly by day 7. Since a boost alone did not induce a detectable titer by day 7, the authors conclude that the E4 boost response is due to cross-reactive memory cells rather than naïve B cells. If this is the case, how do the authors explain the lack of rise in the E3 titer after E4 boosting? Presumably any cross-reactive memory B cells would have been first induced by the E3 prime and therefore should also cross-react with E3. Furthermore, the fusion loop is 100% conserved between the four serotypes of dengue, and previous studies have shown that this antigenic region is immunodominant in the context of a secondary response. Given this, it is surprising that there are apparently few to no cross-reactive memory B cells in the E3 primed mice that are recalled in response to E4 boost. The authors should comment on this unexpected finding. A related point is why slightly different sequence identities between E2 and E3 and E4 and E3 resulted in such big differences. A description of how E3, E2 and E4 differ in their antigenic epitopes might help clarify these points.

3) Previous studies have shown that IgG memory B cells have an increased propensity to differentiate into plasmablasts, whereas IgM memory cells preferentially enter GCs. Did the magnitude of the plasmablast response in the spleen or blood of the immunized mice differ between the E3 boosted mice versus the E2 or E4 boosted mice?

4) It is interesting that the level of VH mutation (Figure 2F) in IgM+ GC B cells on day 7 (post-boost) is lowest in the E4 boosted mice. However, it is unclear whether these IgM+ B cells originate from naïve or memory B cell precursors. How do the average levels of SHM in these cells compare to the day 7 levels of SHM in IgM+ GC B cells of mice that were boosted with E4 in the absence of an E3 prime (i.e. a true naïve response)?

5) The authors show that the VH sequences sampled at day 7 from the E2 boosted mice were biased toward two VH germline genes. Since these VH were also present in the homotypic E3 boost day 7 samples, the authors conclude that these cells likely originate from E3-induced memory B cells. Although these data show that there is significant skewing in the E2 boosted mice, there are very few B cells in the E3 boosted mice that utilize these V-genes (depending on the mouse). For example, there is no VH14-3 usage in mouse#1, that 1 out of 14 VH genes sequenced (joined with J4 and IgHD2-1*01) in mouse #2 and that 2 out of 15 VH genes sequenced (joined with J4 and either with IGHD2-12*01 or IGHD2-14*01) in mouse #3. This sequencing data is not strong enough to support the conclusion that antibodies utilizing these germline genes are derived from cross-reactive memory B cells.

6) Figure 3. The authors claim that the boost with E4 elicited a slower, more "naïve-like" response compared to the E2 boost. However, without showing the responses in a E4 immunized mouse that did not receive an E3-prime (a true naïve response), this conclusion is not strongly supported.

7) In Figure 3 A and B, the competition assay for affinity measurement in E2 or E4 boosted serum for E2 and E4 target, respectively, is complicated and not accurate. The authors should perform antigen-specific ELISAs for the anti-Dengue antibodies, both in the absence and in the presence of urea 7M. This is a well-established assay used to test antibody affinities (Wong-Baeza, et al., Frontiers in Immunology. 2016, Khurana et al., Sci Transl Med, 2011; Puschnik et al., PLoS Negl Trop Dis, 2013; Westerlund et al., Clin Exp Immunol, 2007).

[Editors’ note: what now follows is the decision letter after the authors submitted for further consideration.]

Thank you for submitting your work entitled "Variant proteins stimulate more IgM+ GC B-cells revealing a mechanism of cross-reactive recognition by antibody memory" for consideration by *eLife*. Your article has been reviewed by an original reviewer, and the evaluation has been overseen by a Reviewing and Senior Editor.

Our decision has been reached after consultation between this reviewer and the editors. Based on these discussions and the individual review below, we regret to inform you that your work will not be considered further for publication in *eLife*. In short, we think that two major points raised in the first review were not addressed. Also, we feel that addressing these points will take you a long time, and so it is best to consider publication in another journal.

*Reviewer #2:*

In their revised manuscript the authors made some progress in that they improved some parts, however, they failed to rectify the major weakness present in their main messages, a fault which renders the manuscript unacceptable. Therefore, unfortunately, I am unable to recommend this article in its present form for publication in *eLife*.

Major points:

1) The authors failed to present a clear solution to the comments raised in point 1 and point 4. The authors described that they have not identified a possible contribution of naïve B-cells to the IgM+ GC B-cells they observed after E4 boosting. But at the same time, the authors could also describe that they have not identified a possible contribution of memory B-cells to the IgM+ GC B-cells they observed after E4 boosting.

There is no significant deference between the median level of VH mutation in the primary and secondary response and cross-reactive response to E4. The% of GC B-cells after E4 boosting is slightly high compared to priming, but the authors should pay attention to the previous report indicating that B cells can colonize heterologous GCs and that GCs are dynamic anatomical structures that can be reutilized by newly activated B cells during immune responses (Schwickert et al., J. Exp. Med. 205, 2907).

Thus, the revised manuscript does not give any productive messages to the readers and it does not match the contents presented in the Abstract. As the authors stated in their reply, this issue should be resolved using specificity analysis, at least, by cell transfers, which is perhaps one of the easiest experiments to perform.

2) The authors did respond to the criticisms raised in point 6 and point 7. These tasks improved the manuscript to clearly show that the E4 cross secondary response induces an anti-E4 IgG secondary antibody response. Relative avidity measurement by two different assays (Figure 3B and D) does not match the timing in the avidity maturation in the response, but these results show the affinity maturation in the anti-E4 IgG response, probably in the secondary response.

[Editors' note: further revisions were requested prior to acceptance, as described below.]

Thank you for submitting your article "Variant proteins stimulate more IgM+ GC B-cells revealing a mechanism of cross-reactive recognition by antibody memory" for consideration by *eLife*. The evaluation of your manuscript has been overseen by Arup Chakraborty as the Reviewing and Senior Editor.

The reviewers have discussed the reviews with one another and the Reviewing Editor has drafted this decision to help you prepare a revised submission.

Summary:

The results are of importance for vaccination and affinity maturation, as we have said before.

Essential revisions:

There is only one important issue that needs to be addressed. According to the new data presented in Figure 4F and H, you claim that IgM antibodies from the early E4 boost GC (B7) have higher affinities for E4 than IgM antibodies from the early E4 primary response GC (p7). You assume that IgM antibodies from the early E4 boost GC (B7) are pre-selected by the prior E3 primary immunization, as are memory cells.However, Figure 4H shows that only one IgM antibody showed higher affinity, but other IgMs did not. Can you please provide additional data points to support this conclusion?

---

## [Author Response]

Essential points:1) One major criticism is that the authors are not able to determine whether the IgM+ GC cells are derived from activated naive B cells or from IgM memory B cells. They also do not clone any mAbs from the GC B cells to analyze binding cross-reactivity (only sequence data is shown). The paper would be considerably stronger with this data.

We agree with this criticism and this is an important issue. Whilst such work is a major plan of ours, we consider, however, that the comprehensive level of antibody cloning, and the likely adoptive cell transfers necessary to address these issues, are beyond the scope of this article. We wrote to your journal stating this and you sent the following reply:

“We agree that making the mAbs would take some time. This should be done for a full picture. However if the authors are seeing this as a short report in which there are a number of unanswered questions and in which they lay out the caveats and limitations of their study clearly, and they describe what still needs to be done, that would be sufficient”

In response to this we have inserted the following paragraphs into the Discussion:

“We have not identified a possible contribution of naïve B-cells to the IgM+ GC B-cells we observe after E4 boosting. […] As there are higher proportions of these IgM+ GC B-cells, with fewer mutations relative to the E3 boost, this provides good support for the hypothesis that IgM+ B-cells with fewer mutations furnish memory responses to variant antigens and further challenges the old hypothesis that high affinity, highly mutated, class-switched cells elicited by homotypic antigen boosting are a ‘mirror’ of the antibody memory compartment (Weiss and Rajewsky, 1990)”

2) Figure 1 D and F: The authors show that boosting with E4 did not significantly increase the serum titer to E3 at day 7 or 14. However, the anti-E4 titer rose significantly by day 7. Since a boost alone did not induce a detectable titer by day 7, the authors conclude that the E4 boost response is due to cross-reactive memory cells rather than naïve B cells. If this is the case, how do the authors explain the lack of rise in the E3 titer after E4 boosting? Presumably any cross-reactive memory B cells would have been first induced by the E3 prime and therefore should also cross-react with E3. Furthermore, the fusion loop is 100% conserved between the four serotypes of dengue, and previous studies have shown that this antigenic region is immunodominant in the context of a secondary response. Given this, it is surprising that there are apparently few to no cross-reactive memory B cells in the E3 primed mice that are recalled in response to E4 boost. The authors should comment on this unexpected finding. A related point is why slightly different sequence identities between E2 and E3 and E4 and E3 resulted in such big differences. A description of how E3, E2 and E4 differ in their antigenic epitopes might help clarify these points.

These are important points and we consider that clarification of them has enhanced the article. In the un-revised manuscript results we commented ‘E4 boosting did not significantly increase the anti-E3 titre, even by day 17, showing the E4 variant boost had not induced a significant anti-E3 antibody memory response or the induced antibodies had a low affinity for E3.’

To properly address these points we have added a paragraph to the Discussion:

“Whilst E3 specific memory cells may be expected to increase the anti-E3 titre when stimulated by a cross-reactive E4 boost, the small but not significant effect we observe (Figure 1D) is consistent with the lowest affinity, least mutated, E3-specific memory cells being stimulated by an E4 boost. […] While E2 and E4 are 68% and 63% overall identical to E3, in domain 3, a focus of mouse antibodies, they are 62% and 51% identical, a bigger difference in differences, helping explain the different responses we observe here. Over domains 1 + 2 they are equally different, 82%, from E3.”

3) Previous studies have shown that IgG memory B cells have an increased propensity to differentiate into plasmablasts, whereas IgM memory cells preferentially enter GCs. Did the magnitude of the plasmablast response in the spleen or blood of the immunized mice differ between the E3 boosted mice versus the E2 or E4 boosted mice?

This is a good question. We have not analysed the plasmablast cell response and so cannot directly address this point. Consideration of the serum IgG titres as a proxy for plasmablast levels, however, is informative particularly for comparing the E2 and E4 responses (as there was a low IgG titre against both prior to boosting (unlike with E3)). E2 boosting induced large increases in serum IgG, and relative avidity (new Figure 3A) by day 7. This is consistent with the induction of a strong GC-independent plasmablast response derived from memory cells, many of which are likely to be IgG+ (Dogan et al., 2009; Pape et al., 2011). Contrastingly, the slower and lower serum IgG response to E4 boosting, and the low avidity at day 7 (new Figure 3A) implies a smaller and lower avidity GC-independent plasmablast response, probably due to greater antigenic difference, that is nevertheless much larger than the primary response (see our response to point 2)

We consider it difficult to improve the manuscript to further deal with these issues, and have amended the the second sentence of the Discussion: “All these observations together are consistent with the [E4] response being of an ‘in-between’ type, with lower levels of immediate differentiation of memory cells into antibody forming cells than seen with for example the E2 response.”

4) It is interesting that the level of VH mutation (Figure 2F) in IgM+ GC B cells on day 7 (post-boost) is lowest in the E4 boosted mice. However, it is unclear whether these IgM+ B cells originate from naïve or memory B cell precursors. How do the average levels of SHM in these cells compare to the day 7 levels of SHM in IgM+ GC B cells of mice that were boosted with E4 in the absence of an E3 prime (i.e. a true naïve response)?

We thank you for making this comment. In response we have undertaken an analysis of the primary antibody and GC response to E4. We have added a new figure and section in results to report this. The levels of VH mutation in IgM+ GC B-cells at day 7 after E4 boosting were low anyway (median = 3). Levels of VH mutation 7 days after E4 priming were median = 2, which is lower and suggests that the IgM+ GC B-cells in response to E4 boosting are at least in part memory derived, but this on its own is not conclusive evidence.

As well as the increased levels of mutation in the E4 boost response, the increased numbers of IgM+ GC B-cells and larger more rapid serum IgG boost response (compared to E4 Priming) are evidence for the E4 boost response originating from memory B-cells, but the action of memory T-cells on naïve B-cells cannot be excluded as a contributor. We discuss all these points in the response to point 1, which forms an extra part of the discussion

5) The authors show that the VH sequences sampled at day 7 from the E2 boosted mice were biased toward two VH germline genes. Since these VH were also present in the homotypic E3 boost day 7 samples, the authors conclude that these cells likely originate from E3-induced memory B cells. Although these data show that there is significant skewing in the E2 boosted mice, there are very few B cells in the E3 boosted mice that utilize these V-genes (depending on the mouse). For example, there is no VH14-3 usage in mouse#1, that 1 out of 14 VH genes sequenced (joined with J4 and IgHD2-1*01) in mouse #2 and that 2 out of 15 VH genes sequenced (joined with J4 and either with IGHD2-12*01 or IGHD2-14*01) in mouse #3. This sequencing data is not strong enough to support the conclusion that antibodies utilizing these germline genes are derived from cross-reactive memory B cells.

We agree with this point. There are not enough VH14-3 and VH14-4 sequences in the E3 boost day 7 sample to argue that the large numbers of these sequences after E2 boosting is evidence of a cross reactive response involving these V-genes, and we have amended the text in the Results and Discussion to remove this implication, even though considering VH14-3 and VH14-4 after E3 boosting there are 2 in mouse 2, and 3 in mouse 3. We consider that the large numbers of VH14-3/4 sequences present after the E2 boost is evidence of a secondary response that is somewhat focused on a particular epitope, and have inserted a sentence in the results to this effect instead. Cross reactive secondary responses focusing on particular epitopes have been identified in humans and are further discussed in the response to point 2 of the reviewers comments, and form part of an addition to the discussion

6) Figure 3. The authors claim that the boost with E4 elicited a slower, more "naïve-like" response compared to the E2 boost. However, without showing the responses in a E4 immunized mouse that did not receive an E3-prime (a true naïve response), this conclusion is not strongly supported.

We take this point, as it refers to Figure 3, to mean the use of the statement “naïve-like” to describe the rise in affinity in response to E4 boosting. We agree that this use is incorrect in this context as we had not done an analysis of a primary/naïve response. We have now performed an analysis of the primary response to E4 and have added this to the Results section accordingly. These results show that the real naïve/primary serum IgG response to E4 is at a much lower titre compared to boosting as might be expected. Titres were too low to perform a relative affinity assay and no 7M Urea resistant IgG antibodies were detected at day 7 or 17 after prime (data not shown). We have removed the phrase referred to (naïve-like) from the serum avidity/affinity section of the results

7) In Figure 3 A and B, the competition assay for affinity measurement in E2 or E4 boosted serum for E2 and E4 target, respectively, is complicated and not accurate. The authors should perform antigen-specific ELISAs for the anti-Dengue antibodies, both in the absence and in the presence of urea 7M. This is a well-established assay used to test antibody affinities (Wong-Baeza, et al., Frontiers in Immunology, 2016, Khurana et al., Sci Transl Med, 2011; Puschnik et al, PLoS Negl Trop Dis, 2013; Westerlund et al, Clin Exp Immunol, 2007).

This is a very valuable point and we have performed a relative avidity assay/7M Urea resistance assay on the serum samples that enhances this study.

We consider, however, that these two assays measure different things. Resistance to 7M Urea is measuring the avidity of a divalent binding of antibody to immobilized target. The competition assay we have used is more a measure of relative affinity as it is measuring the ability of monovalent interactions between antibody and E-protein in solution to inhibit the binding of the antibody to the immobilized E-protein target.

For this reason, and because doubts about the quality of the Urea resistance assay have recently been raised (Alexander et al., 2015), particularly with respect to antibodies against conformational epitopes, which are present on E-proteins, we have included both assays in the revised article and modified the Results section accordingly.

Readings for the 2 assays correlate but show what looks like a discrepancy too, although there is an explanation:

After E2 boosting there is a rapid increase in serum avidity by day 7 detected by the Urea assay and no apparent further increase until day 32 perhaps, although data variability is high at day 7 and 17. The affinity for E2 measured by competition assay, however, increases from day 7 to day 17. We interpret this to mean that only a small portion of serum IgG has undergone affinity maturation by day 17, which is plausible because of the high levels of serum IgG already present, so it would not be detected by the Urea assay, particularly with the high data variability seen with this assay at these time points, but would by the competition assay. If this were the case (that a minor portion of serum IgG had affinity matured) the competition assay may discriminate less between E2 boost day 7 and day 17 samples at the highest competitor concentrations and this is what we observe. Another explanation for the lack of detectable improvement in avidity for E2 between day 7 and day 17 is that some or all of the affinity matured IgG is against conformational epitopes vulnerable to 7M Urea.

Both assays show a slower, but equivalent, if not greater, increase in affinity/avidity for E4 after E4 boost compared to affinity/avidity for E2 after E2 boost, which is an important observation.

As well as revising the Results section to include the new data, we have also amended the first line of the Discussion to: ‘The most variant protein we immunized with, E4, stimulated GCs with the highest proportion of IgM+ cells with the lowest levels of VH gene mutation, greater VH-gene diversity, and a slower, more specific, serum IgG response that interestingly eventually resulted in equivalent if not higher affinity compared to the E2 boost’

[Editors' note: the author responses to the re-review follow.]

Reviewer #2:

*In their revised manuscript the authors made some progress in that they improved some parts, however, they failed to rectify the major weakness present in their main messages, a fault which renders the manuscript unacceptable. Therefore, unfortunately, I am unable to recommend this article in its present form for publication in* eLife.Major points:1) The authors failed to present a clear solution to the comments raised in point 1 and point 4. The authors described that they have not identified a possible contribution of naïve B-cells to the IgM+ GC B-cells they observed after E4 boosting. But at the same time, the authors could also describe that they have not identified a possible contribution of memory B-cells to the IgM+ GC B-cells they observed after E4 boosting.There is no significant deference between the median level of VH mutation in the primary and secondary response and cross-reactive response to E4. The% of GC B-cells after E4 boosting is slightly high compared to priming, but the authors should pay attention to the previous report indicating that B cells can colonize heterologous GCs and that GCs are dynamic anatomical structures that can be reutilized by newly activated B cells during immune responses (Schwickert et al., J. Exp. Med. 205, 2907).Thus, the revised manuscript does not give any productive messages to the readers and it does not match the contents presented in the Abstract. As the authors stated in their reply, this issue should be resolved using specificity analysis, at least, by cell transfers, which is perhaps one of the easiest experiments to perform.

In Revision 1 we attempted to address point 4 by analysing SHM levels as suggested but the results were inconclusive.

Both these points could be satisfactorily addressed by the analysis and comparison of recombinant antibodies from GC B-cells after E4 priming and after E4 boosting (after E3 priming), which we have now done.

With our new data we show that IgM antibodies from the early E4 boost GC have higher affinities for E4 than IgM antibodies from the early E4 primary response GC, implying they have been pre-selected by the prior E3 primary immunisation, and so are memory cells. This provides further and clear evidence for the main proposal of the study, and much improves it as suggested by the referee above.

We are reporting about antibodies from B-cells, which have not been selected for antigen binding in vitro, from early in the response. Little is known about these. Whilst we measure clear differences in rAb affinity, overall the affinities are low, as would be expected from early GC. Analysis of specificity shows that the rAbs recognise E3 and E4 similarly, even from E4 primed mice. We suggest that at these affinities antibodies can’t discriminate between similar epitopes. This provides a starting point to debate the meaning of ‘cross-reactivity’ in antibody memory, of importance for the understanding of for example the generation of bnAbs against HIV, the germline precursors of which have similar affinities to our rAbs.

[Editors' note: further revisions were requested prior to acceptance, as described below.]

Essential revisions:There is only one important issue that needs to be addressed. According to the new data presented in Figure 4F and H, you claim that IgM antibodies from the early E4 boost GC (B7) have higher affinities for E4 than IgM antibodies from the early E4 primary response GC (p7). You assume that IgM antibodies from the early E4 boost GC (B7) are pre-selected by the prior E3 primary immunization, as are memory cells.However, Figure 4H shows that only one IgM antibody showed higher affinity, but other IgMs did not. Can you please provide additional data points to support this conclusion?

We have previously submitted the second revision, but some issues came to light regarding the clarity of the data presentation in Figure 4, as this figure contained panels with data points and labels that were not fully explained in the main text and figure legend.

We have now amended the main text and relevant figure legend to clarify this matter.